# Household survey on owned dog population and rabies knowledge in selected municipalities in Bulacan, Philippines: A cross-sectional study

Timothy John R. Dizon[1][ʘ]*, Nobuo Saito[2,3][ʘ], Marianette Inobaya[1], Alvin Tan[1], Mark Donald C. Reñosa[1], Thea Andrea Bravo[1], Vivienne Endoma[1], Catherine Silvestre[1], Micah Angela O. Salunga[2], Patricia Mae T. Lacanilao[2], Jerric Rhazel Guevarra[1], Yasuhiko Kamiya[3], Maria Glofezita O. Lagayan[3,4], Kazunori Kimitsuki[2], Akira Nishizono[2], Beatriz P. Quiambao[1]

**1** Research Institute for Tropical Medicine, Muntinlupa City, Metro Manila, Philippines, **2** Department of Microbiology, Oita University Faculty of Medicine, Yufu, Oita, Japan, **3** School of Tropical Medicine & Global Health, Nagasaki University, Nagasaki, Nagasaki, Japan, **4** Bureau of Animal Industry, Department of Agriculture, Quezon City, Metro Manila, Philippines

ʘ These authors contributed equally to this work.
* timothy.dizon@ritm.gov.ph

**Data Availability Statement:** All relevant data are within the manuscript and its Supporting Information files.

## Abstract

### Background

Despite the effort to eradicate rabies in the Philippines, human rabies cases have not decreased in the past decade. Rabid dogs pose the most significant hazard in the countries with the highest burden of rabies, and 70% rabies vaccine coverage is recommended for dogs in high-risk areas. Ascertaining the owned dog population and community knowledge on rabies can help improve vaccine coverage and information campaigns.

### Methodology/Principal findings

We conducted a cross-sectional survey in six randomly selected communities (five urban, one rural) in Central Luzon, Philippines. We first conducted the complete mapping of 9,173 households and then randomly selected 727 households. More than half (54.1%) of the households owned dogs (1.21 dogs/household). In the 727 households, we identified 878 owned dogs and 3256 humans. According to these results, the dog-to-human ratio was approximately 1:3.7. Only 8.8% of households reported a history of dog bite in 2019. Among dog-owning households, 31% reported that they allow their dogs to roam freely. Of the recorded dogs, 35.9% have never been vaccinated, and only 3.5% were spayed or castrated. Factors associated with lower rabies knowledge include (1) no education aOR: 0.30 (0.16–0.59), and (2) only primary school education aOR: 0.33 (0.22–0.49). In contrast, factors associated with higher knowledge include (1) owning a dog and not allowing them to roam freely aOR: 2.01 (1.41–2.87) and (2) owning a dog and allowing them to roam freely aOR: 1.84 (1.17–2.92), when compared to those with no dogs.

**Funding:** This work was supported by a JICA/ AMED SATREPS (Science and Technology Research Partnership for Sustainable Development)(https://www.jst.go.jp/global/english/ index.html) for "The establishment of the one health prevention and treatment network model for the elimination of rabies in the Philippines" (No.17823721) to AN. The funders had no role in study design, data collection and analysis, decision to publish, or preparation of the manuscript.

**Competing interests:** The authors have declared that no competing interests exist.

## Conclusions/Significance

We identified a larger dog population in the community than the usual estimates (1:10), suggesting that annual vaccine needs in the Philippines must be reassessed. Our survey shows a relatively good understanding of rabies; however, awareness of the concept of rabies as a disease, and how animals and humans can acquire it, is lacking.

### Author summary

Rabies is a fatal disease primarily transmitted by rabid dogs. It is estimated that 59 000 human deaths occur worldwide annually because of rabies. Prevention is possible using pre-exposure prophylaxis or post-exposure prophylaxis (PEP). Despite this, rabies remains a neglected disease and a burden, particularly in developing countries. In this study, we found that the current estimated dog to human ratio of 1:10 may be incorrect. Our data show that the ratio of 1:3.7 may more closely reflect real-world figures. This finding is supported by previous studies on dog populations. We provide compelling evidence that there is a need to revise the current guidelines for estimating the dog population to provide adequate vaccine coverage. Our findings on community knowledge about rabies show that there is relatively good understanding regarding rabies; however, awareness of the concept of rabies as a disease and how animals and humans can acquire it is lacking. This should be a point of focus for future education campaigns.

## Introduction

Rabies is an acute, progressive viral infection that is nearly always fatal. Rabid dogs pose the most significant hazard in the countries with the highest burden of rabies. It is estimated that 59 000 human deaths occur worldwide annually because of rabies, and 99% of cases are transmitted from dogs [1]. The majority of cases, 95%, occur primarily in Asia and Africa, with these continents experiencing 35 172 and 21 476 deaths per year, respectively [2]. Rabies is a zoonotic disease that can be prevented through a combination of different measures, such as raising awareness through education campaigns, responsible pet ownership, enforcement of animal vaccination policies, eliminating exposure to rabid animals, and provision of post-exposure prophylaxis (PEP) [3]. However, despite considerable scientific progress, rabies is a neglected disease and presents a modern public health problem, particularly in developing countries [3,4].

Through the Anti-Rabies Act of 2007, the Philippine government has taken steps to reduce the burden of this disease [5,6]. The Philippines is one of the leading countries that has built an expansive decentralized network of Animal Bite Treatment Centers (ABTC) to provide patients with free or low-cost PEP for rabies [7]. The number of ABTCs in the country have progressively increased through the years. Data from the health department show that the number of ABTCs grew from 227 in 2005 to 384, 513 and 613 in 2011, 2017 and 2018 respectively [6,8,9]. The government instituted free full PEP courses in the 408 ABTCs throughout the country, starting in 2016 [10]. This was a shift from the previous policy which charged victims for two of the four doses required for PEP [11]. There is also an increasing trend in the number of reported animal bites and, by extension, PEP given.

Data from the health department show that there were approximately 1.1 million (1 159 711) reported animal bite cases in 2018 [6]. This is an increase from the reported figure of 1.089 million in 2016 [11], 330 077 in 2011 and 206 253 in 2009 [8]. The huge jump in reported bite cases in 2016 may be attributed to the change of policy described earlier, covering the whole cost of PEP. Despite the enormous effort to provide bite victims with PEP, no apparent downward trend in human rabies cases has been observed in the past decade (2008–2018). The average number of annual human rabies cases in the Philippines between 2008 and 2018 was 231 (range, 205–276) [6,11]. Although PEP can prevent human deaths, it has little impact on transmission of the rabies virus among domestic dogs. Mass animal vaccination is the most effective method for controlling rabies in the community. Sustainable vaccination of 70% of the at-risk dog population is a critical factor [12]. However, without an accurate estimate of the dog population, it would be very difficult to provide an adequate number of animal vaccines and to determine the staffing and budget necessary for a successful vaccination campaign. In many areas of the Philippines, owned dog registration is poorly implemented. If the local dog population data are unavailable, they are estimated using the 1:10 dog to human ratio, recommended by the national guidelines [6]. Several previous studies attempted to ascertain the dog: human ratio in the Philippines and showed gaps between their survey results and estimation using the 1:10 ratio [13–17]. Thus, underestimating the dog population would lead to insufficient vaccine coverage. The National Rabies Prevention and Control Program (NRPCP) recommendation has not been changed because of insufficient dog population data. Moreover, dog population data in highly populated and crowded areas are still inadequate. Further studies to estimate the dog population, particularly in areas with high rabies endemicity, are necessary.

Other hallmarks of rabies control include raising awareness through education campaigns and responsible pet ownership. Previous studies in the Philippines about rabies knowledge, attitude, and practices (KAP) have highlighted commonly held misconceptions, including beliefs in traditional healers and incorrect bite wound treatments such as garlic and vinegar application [18]. Another study showed that the knowledge of pet owners regarding the existence of the Anti-Rabies Act and its penalties for non-compliance is poor [5]. Another study revealed that lack of awareness and insufficient funds are the primary reasons for not visiting the ABTC [7]. Despite the identified knowledge gaps, there is a consensus among the rabies KAP studies conducted in the Philippines that the population has a relatively good knowledge about rabies and responsible pet ownership [15,19,20].

We conducted the study with two major objectives. The first was a dog ecology and owned dog population survey, and the second was to ascertain community knowledge on rabies. Given the recommendation of the rabies control program of using a 1:10 ratio when dog population numbers are unavailable, we hypothesized that identifying gaps in owned dog population could provide better estimates and support vaccination strategies in the future. Previous KAP studies have a consensus that knowledge about rabies is relatively good, however we hypothesized that ascertaining weaknesses in rabies knowledge could be a focal point for developing future targeted education campaigns.

A point of concern over the results of the previous studies is related to the high population density in some areas of the Philippines. Often communities have limited household registration and have dense arrangement of households. The use of cluster, purposive, and systematic random sampling to select survey households might lead to some underrepresented households and introduce bias [13–17,21]. This study is unique among its peers as we conducted complete mapping of the households in the target communities and systematically selected households for survey. Additionally, this was done in a densely populated province in central Luzon where the incidence of human rabies is high.

## Methods

### Ethics statement

Ethical approval was obtained from the Research Institute for Tropical Medicine, Institutional Review Board (RITM-IRB), Philippines (IRB protocol number 2019–31). We also obtained approval from the RITM Institutional Animal Care and Use Committee (ID number 2019–08). Trained staff sought and obtained written informed consent from all participants before the survey.

### Study site

We conducted this cross-sectional study in six municipalities of the province of Bulacan in the Philippines, namely Bulakan, Calumpit, Guiguinto, Hagonoy, Paombong, and Pulilan (Fig 1), with the selected households highlighted (Fig 2). Bulacan is in the Central Luzon Region (Region III), north of the National Capital Region. This province was chosen because human rabies cases have been consistently reported there, with a total of 32 human rabies cases reported from 2015 to 2018 (incidence rate, 0.97/100 000 population, median: 8, range: 5–11) and a total of 134 animal rabies cases between 2015 and 2018 (median: 32.5, range: 25–44) (S3 Table). Bulacan has a land area of 262 500 hectares and comprises 21 municipalities, 3 component cities, and 569 barangays [22]. In the Philippines, a barangay is the smallest territorial and administrative level of the government. According to the 2015 census, Bulacan had a population of 3 292 071; the total population of the selected municipalities was 576 910. Using the established dog to human ratio of 1:10, the dog population estimated for Bulacan was 329 207 with 57 691 in the selected municipalities. Many households are remarkably crowded; one

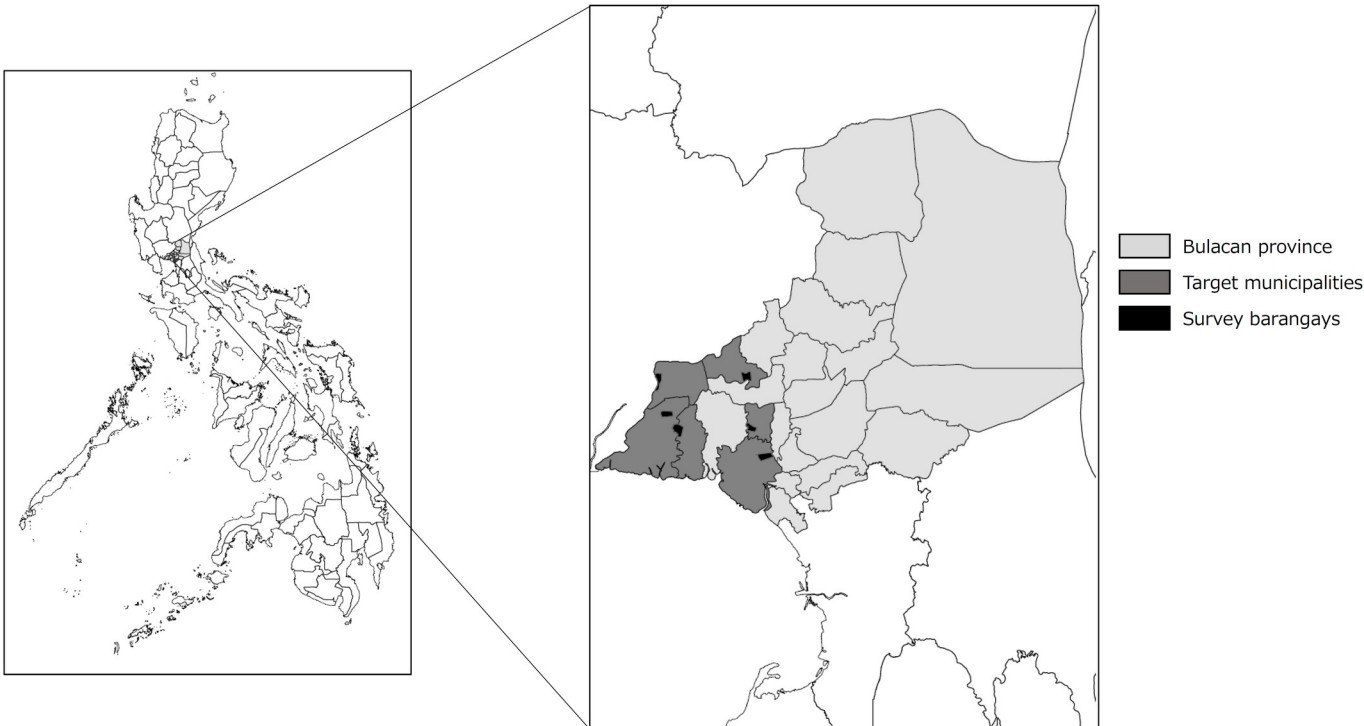

**Fig 1. Maps Indicating the Location of Bulacan in the Philippines.** Bulacan (gray), target municipalities (dark gray), selected municipalities (black). Provincial and municipal boundary data were taken from the United Nations Office for the Coordination of Human Affairs (OCHA). (https://data.humdata.org/dataset/philippines-administrative-levels-0-to-3).

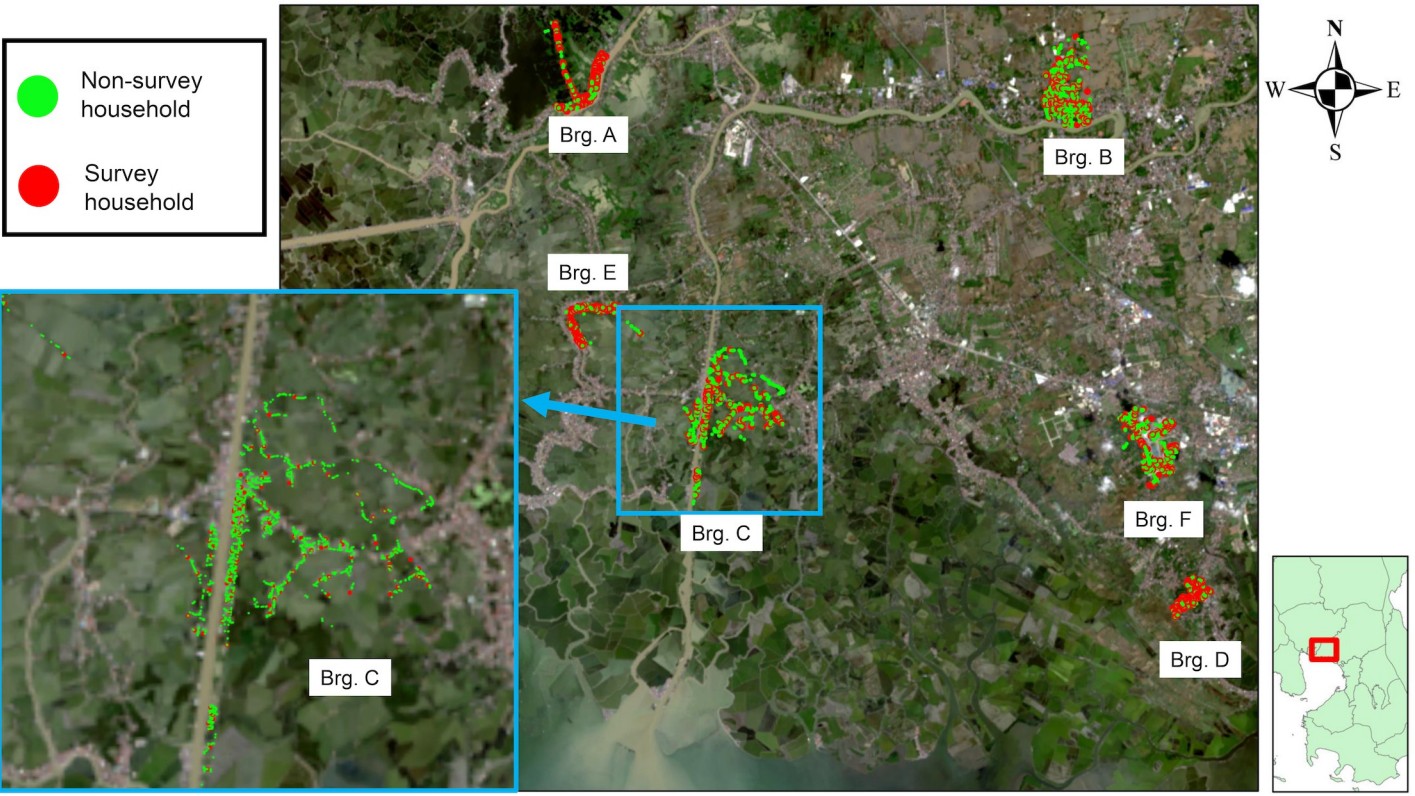

**Fig 2. Maps Indicating the Locations of the Selected Households in the 6 Barangays.** Abbreviations: Brg, Barangay. Survey households (red dots) and non-survey households (green dots). Number of survey households /Total number of households: Brg A (127/883), Brg B (118/2109), Brg C (116/2213), Brg D (126/820), Brg E (123/1133), Brg F (117/2018). Base maps were taken from U.S. Geological Survey (USGS)—All maps are in the public domain. (https://earthexplorer.usgs.gov/scene/metadata/full/5e83d0b656b77cf3/LC81160502020359LGN00/).

parcel of land is divided into multiple portions, with most houses located one behind the other. Research staff can access these households through small paths, usually within many boundaries. Therefore, a simple selection of the households may not provide precise randomization and can cause bias.

## Sample size calculation and household sampling procedure

Since this is a baseline survey for community-based information education and communication campaign, in which the primary outcome is not allowing dogs to roam freely. The sample size was estimated using a survey by Barroga et al. done in Bicol in 2015. In this survey, not allowing dogs to roam freely was estimated to be practiced by 28.5% of households with dogs [23]. Calculations were done using a power of 80% and a design effect of 2.0 for the multi-stage stratified systematic sampling that will be done. We assumed that selecting 689 households would provide adequate information (>80%) to study the proportion of households with dogs, considering a 35% adjustment, and taking missing information and refusal to provide information into account.

For each municipality, one barangay was randomly selected with a probability of selection proportional to the number of households in the barangay. The number of households was estimated by dividing the barangay population based on the 2015 census by a factor of 4.4, the average household size in the Philippines in 2015. Data from the Philippine Statistics Authority showed that five of the six randomly selected barangays were classified as urban. Only

barangay A is rural [24]. We obtained the geographic information system coordinates of all households in the target barangays from January 15 to 31, 2020. Trained staff visited all the households and logged their global positioning system coordinates using Garmin eTrex 10 (Kansas City, Missouri, USA), and assigned a unique ID to each. The coordinates were loaded as KML files, and households were randomly selected using Microsoft Excel based on each barangay's determined sampling interval. If the residents of a prospective household were not available on the first visit, two follow-up visits were made. If the residents were still unavailable after the third visit or refused to participate, the next higher-numbered household was recruited. If the replacement household did not participate, the next lower-numbered household of the original sampled household was recruited. This was repeated until a replacement was recruited, or the two possible replacements were exhausted.

## Data collection

A face-to-face survey was conducted from February 4 to 19, 2020. The participants were interviewed using a structured questionnaire that was developed based on previous studies on rabies knowledge and dog ecology [7,25]. Our data collectors requested an interview with the household head or other adults present if the household head was unavailable. Two common languages, English and Filipino, were used in this survey. We collected the demographic information of the participants, data on animal bite history, and details on knowledge and practices about rabies. Participants who reported that they had heard of rabies were questioned on rabies knowledge, while those who reported a history of dog bite in 2019 were asked questions on practices. If the selected households had dogs, additional variables on dog ownership and population were also collected. In this study we used the definition indicated in the World Organization for Animal Health (OIE) Terrestrial Code which "means a dog for which a person claims responsibility" [26]. This definition would include both restrained and unrestrained owned dogs. Study data were collected using Kobo Collect (version 1.25.1) installed on an Android smartphone and managed using Kobo Toolbox (Cambridge, Massachusetts, USA, version 2.019.42). Data were sent from the smartphones to the Kobo Toolbox server after each survey day. Range, logical checks, and skipping logic were integrated into the electronic form. The survey questions are available as a Microsoft Excel file using the XLSForm standard and can be used in Kobo Collect and Kobo Toolbox (S1 File) [27].

## Data analysis

Survey data were exported from the Kobo Toolbox server into Microsoft Excel (Redmond, WA, USA) and analyzed using Stata Statistical Software, version 15.1 (StataCorp, College Station, TX). Percentages were computed for all applicable variables to describe the respondents' demographic profiles, KAP, and owned dog information. The owned dog population in each barangay was estimated by determining the dog: human ratio based on the data gathered from this survey. This was done by dividing the total population of dogs by the total number of humans determined through the survey. The ratio was then used on the 2015 census data of each municipality to estimate the total number of owned dogs. The number of vaccinated dogs through the rabies vaccination program of the government was obtained from the Provincial Veterinary Office (PVO) records. This did not include vaccinations done outside the program. Using the vaccination data and dog population estimates, we determined the partial vaccination coverage of the selected municipalities. We mapped all households using GIS software (ArcGIS version 10.5; ESRI). Population data and basic maps were obtained from the Philippines Statistics Authority, United Nations Office for the Coordination of Human Affairs (OCHA) and the United States Geological Survey (USGS).

To assess the knowledge and perception of the respondents regarding rabies, we created a scoring system based on nine questions that we determined were essential knowledge queries about rabies (S1 Table). Participants could obtain a maximum overall score of 41 and a minimum of 0. If the participant scored higher than the average score, we categorized them into the high-score group. We performed logistic regression analysis to determine the socio-demographic factors associated with higher scores (binary outcomes). We included all significant variables from the univariate analysis in a multivariable logistic regression model using a backward stepwise approach. We retained significant variables and included age and sex in the final models because they were considered confounding factors. The Hosmer-Lemeshow test was used to check for goodness of fit.

## Results

### Socio-demographic characteristics of surveyed households

The socio-demographic characteristics of the surveyed households are shown in Table 1. We identified 9173 households in the six barangays and 727 households were randomly selected and surveyed. All participants were able to complete the survey. The total number of household members was 3256, with a mean household size of 4.5 (95% confidence interval [CI]: 4.3–4.6). Almost half of the surveyed households (n = 298, 41.0%) earn between 50 000 and 150 000 Philippine Pesos (PHP) (1 000 and 3 000 USD) per year, while 21.5% (n = 156) earn between 151 000 and 250 000 PHP (3 020 and 5 000 USD) per year. Most of the respondents (n = 516, 71.0/%) had completed primary and secondary education. Only a few households (n = 64, 8.8%) reported that they had at least one person who was bitten by a dog in 2019, with the number of bite victims determined to be 84 or 2.6% of the total surveyed population. Regarding allowing the dog to roam freely, our data show that economic status and household size were not statistically associated with this practice. However, there was a statistically significant association between education level and this practice (p = 0.03), with lower education leading to higher practice of this behavior. There was also a statistically significant difference regarding this practice among the barangays (p<0.01). The household characteristic of a family member having a history of dog bite was statistically associated with household size (p<0.01). Economic status and education were not statistically associated with this household characteristic.

### Dog ownership, dog ownership practices, and estimation of the owned dog population

More than half of the surveyed households (54.1%, n = 393) had dogs, with 44% (n = 173) having one dog, 39.4% (n = 155) 2–3 dogs, 11.5% (n = 45) 4–5 dogs, and 5.1% (n = 20) 6–10 dogs. Most respondents (65.4%) said they kept dogs to act as guard dogs. Some were the offspring of a previously owned dog (28.1%), while others were either purchased (11.3%), adopted (5.9%) or received as a gift (54.7%). The average number of dogs per household was 1.21 (95% CI: 1.09–1.33). The majority of households (n = 271, 69.0%) reported that they did not allow their dogs to roam, while a certain number (n = 122: 31.0%) allowed their dogs to roam freely outside the house.

We recorded a total number of 878 owned dogs among survey households. Among them, 224 (25.5%) were allowed to roam freely. Further, 316 owned dogs (35.9%) had never been vaccinated, and only 31 (3.5%) were spayed or castrated. The owned dog population survey data shows that the average dog population per household was 1.21 (95% CI: 1.09–1.33), and the dog population per human population was 0.27 (95% CI: 0.25–0.29).

**Table 1. Socio-demographic Characteristics of Households That Participated in the Study.**

| | Total N (%) | Own dog N (%) | P-value | Households allowing at least one dog to roam freely N (%) | P-value | Has anybody living in this household been bitten by a dog in 2019? (Jan—Dec 2019) | |
|---|---|---|---|---|---|---|---|
| Total | 727 | 393 (54.1) | | 122 (31.0) | | 64 (8.8) | |
| Income (PHP/month) | | | | | | | |
| Less than 50 000 | 84 (11.6) | 50 (59.5) | 0.16 | 16 (32.0) | 0.28 | 6 (7.2) | 0.12 |
| 50 000–150 000 | 298 (41.0) | 146 (49.0) | | 50 (34.3) | | 32 (10.7) | |
| 151 000–250 000 | 156 (21.5) | 90 (57.7) | | 31 (34.4) | | 16 (10.3) | |
| more than 251 000 | 94 (12.9) | 57 (60.6) | | 11 (19.3) | | 2 (2.1) | |
| Unknown or declined to answer | 95 (13.1) | 50 (52.6) | | 14 (28.0) | | 8 (8.4) | |
| Education status of respondents** | | | | | | | |
| None | 58 (8.0) | 32 (55.2) | 0.03 | 14 (43.8) | 0.03 | 6 (10.3) | 0.91 |
| Primary education | 220 (30.3) | 113 (51.4) | | 43 (38.1) | | 19 (8.6) | |
| Secondary education | 296 (40.8) | 149 (50.3) | | 43 (28.9) | | 24 (8.1) | |
| Post-secondary education | 152 (20.9) | 98 (64.5) | | 22 (22.5) | | 15 (9.9) | |
| Household size | | | | | | | |
| Less than 3 | 245 (33.7) | 122 (49.8) | 0.18 | 36 (29.5) | 0.72 | 16 (6.5) | <0.01 |
| 4–6 | 370 (50.9) | 204 (55.1) | | 67 (32.8) | | 26 (7.0) | |
| more than 7 | 112 (15.4) | 67 (59.8) | | 19 (28.4) | | 22 (19.8) | |
| Municipality | | | | | | | |
| Barangay A, Calumpit | 127 (17.5) | 72 (56.7) | 0.09 | 18 (25.0) | <0.01 | 15 (11.9) | 0.39 |
| Barangay B, Pulilan | 118 (16.2) | 66 (55.9) | | 27 (40.9) | | 8 (6.8) | |
| Barangay C, Paombong | 116 (16.0) | 71 (61.2) | | 34 (47.9) | | 9 (7.8) | |
| Barangay D, Bulakan | 126 (17.3) | 71 (56.4) | | 17 (23.9) | | 14 (11.1) | |
| Barangay E, Hagonoy | 123 (16.9) | 63 (51.2) | | 13(20.6) | | 12 (9.8) | |
| Barangay F, Guiguinto | 117 (16.1) | 50 (42.7) | | 13 (26.0) | | 6 (5.1) | |

* 1 missing value

** 3 missing values

The average dog: human ratio among the barangays, based on the survey data, was 1:3.7. The estimated owned dog population based on the 1:3.7 ratio was almost three times the number estimated using the 1:10 ratio (152 484 vs. 56 548 dogs). In 2018, 23 584 dogs were vaccinated in the study municipalities based on the PVO data. The vaccine coverage estimated using the 1:10 ratio was 41.7%, while that estimated using the 1:3.7 ratio was 15.5%. The data on dog ownership, dog ownership practices, whether they allowed their dogs to roam freely, and estimation of the owned dog population and vaccine coverage are listed in Table 2.

**Table 2. Dog Population, Ownership Characteristics, Estimated Dog Population, and Estimated Dog Vaccine Coverage of Surveyed Households in Bulacan.**

| | | N (% or 95% CI) |
|---|---|---|
| Total number of households surveyed | | 727 |
| Total members of the surveyed households | | 3256 (95% CI: 3145–3366) |
| Average number of people per household (Mean) | | 4.5 (95% CI: 4.3–4.6) |
| Household owning at least one dog | No | 334 (45.9%) |
| | Yes | 393 (54.1%) |
| Number of dogs owned per household | 1 dog | 173 (44.0%) |
| | 2–3 dogs | 155 (39.4%) |
| | 4–5 dogs | 45 (11.5%) |
| | 6–10 dogs | 20 (5.1%) |
| Number of households allowing at least one dog to roam freely | | |
| | No | 271 (69.0%) |
| | Yes | 122 (31.0) |
| Total number of dogs in surveyed households | | 878 (95% CI 814–942) |
| Total number of freely roaming dogs | | 224 (25.5%, 95% CI 198–249) |
| Total number of dogs that were never vaccinated | | 316 (35.9%, 95% CI 288–344) |
| Total number of dogs that were spayed or castrated | | 31 (3.5%, 95% CI 20–42) |
| Average number of dogs per household (Mean) | | 1.21 (95% CI: 1.09–1.33) |
| Average dog population per human (Mean) | | 0.27 (95% CI: 0.25–0.29) |
| Dog: human ratio | | 1:3.7 (95% CI 3.4–4.0) |
| Estimated dog population of 6 municipalities (human population, 565 476) using the dog: human ratio of 1:3.7 | | 152, 484 (95% CI: 141, 369–163, 988) |
| Estimated vaccine coverage in 2018 using dog: human ratio of 1:3.7 | | 15.5 (95% CI: 14.4–16.7) |
| Estimated vaccine coverage in 2018 using dog: human ratio of 1:10 | | 41.7% |

CI, confidence interval

## Knowledge regarding rabies

The knowledge of rabies among the surveyed households is shown in Fig 3. Of the 727 respondents, 85.4% (n = 621) claimed to have heard of rabies. Our data showed that the primary source of information for the respondents was television (40.6%, n = 252) followed by neighbors and relatives (33.0%, n = 205). Other sources of information that remained underutilized were the internet and social media (16%, n = 99) and other community stakeholders such as health workers and veterinarians (14.8%, n = 92). Almost three-quarters (71.66%, n = 521) of the households knew the ABTC near their place.

Humans (77.03%), dogs (72.08%), and cats (61.35%) were identified as vulnerable to rabies. The respondents identified salivating or drooling as the most common sign of rabies in dogs (53.8%), while fear of water was identified as the most common sign of rabies in humans (21.1%).

Regarding knowledge of rabies pathogenesis, more than half of the respondents (57.22%) answered that they could get rabies if dogs bit them, half of the respondents (49.9%) answered that dogs could get rabies by eating dirty food, while a quarter (20.36%) answered that rabid dogs could transmit rabies. Answers to questions on knowledge of rabies prevention revealed that almost half of the respondents knew that not allowing dogs to roam freely can prevent rabies (44.4%), while some (22.7%) considered vaccination a preventive measure. However, on asking how frequently dogs should be vaccinated for rabies, more than three-fourths of the respondents answered incorrectly or did not know the answer (78%). More than half of the

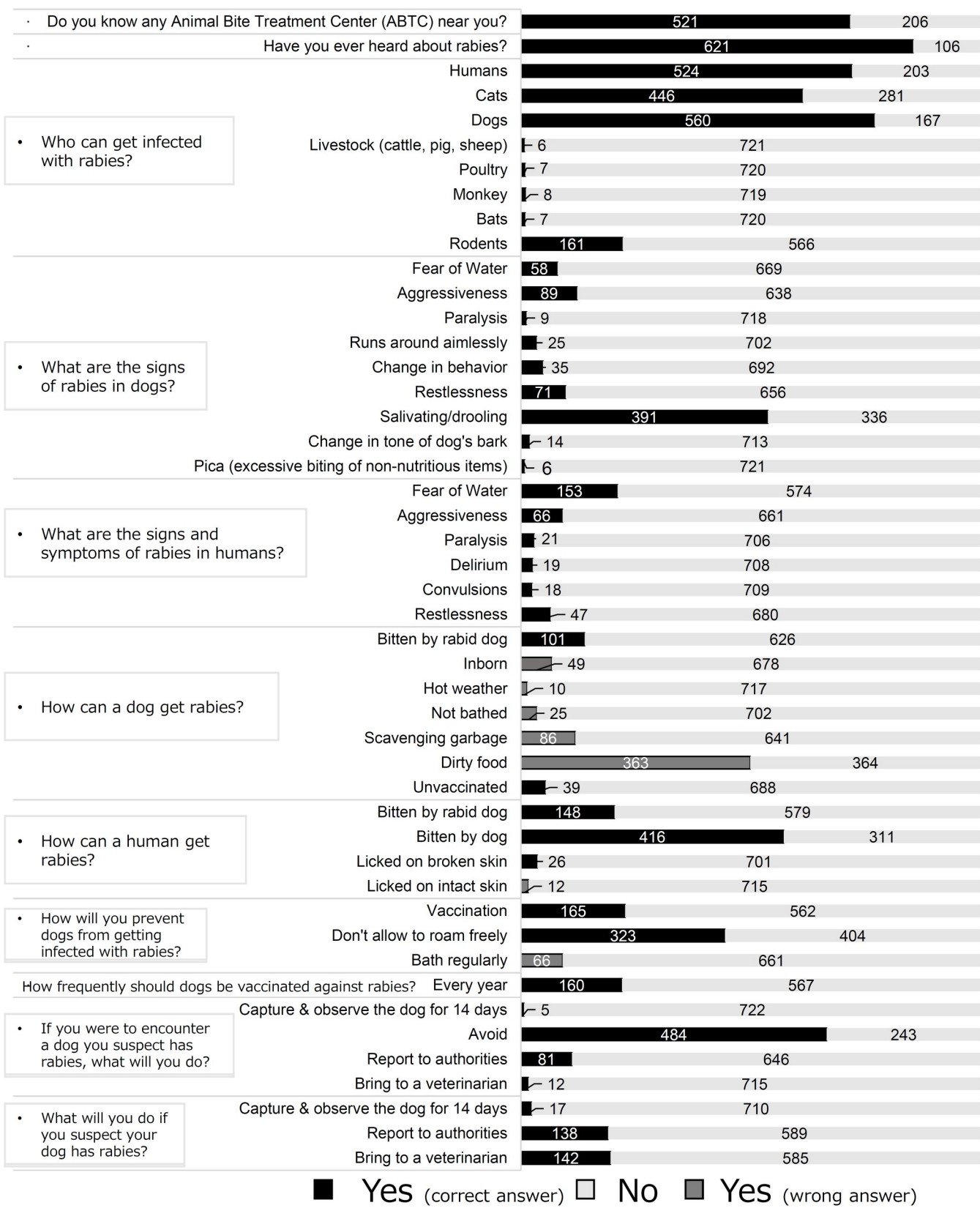

**Fig 3. Knowledge of Rabies Among Surveyed Households.**

respondents (66.6%) said they would avoid dogs that they suspect of having rabies, and 11.14% said they would report dogs suspected of having rabies to the authorities. When asked about their course of action if they suspected their dog had rabies, 19.5% said they would take it to a veterinarian, and 19% said they would report it to the authorities.

## Rabies knowledge and perception score

The univariate and multivariate analyses for rabies knowledge and perception scores are presented in Table 3. The mean score was 10.8 (range: 0–23), and the cut-off was 11. If the participant scored ≥11, they were considered to have "good" knowledge and perception of rabies, and if they scored lower, they were considered to have "poor" knowledge and perception of rabies. No education (adjusted odds ratio [aOR]: 0.30, 95% CI: 0.16–0.59) or only primary school education (aOR: 0.33, 95% CI: 0.22–0.49) were significantly associated with a lower score compared to secondary school education. There was a significant difference in scoring among the barangays, with respondents from barangay F scoring higher than those from barangay A (aOR: 2.92, 95% CI: 1:68–5.06). Owning a dog and not allowing them to roam freely aOR: 2.01 (1.41–2.87) and owning a dog and allowing them to roam freely aOR: 1.84 (1.17–2.92) are factors associated with higher knowledge when compared to those with no dogs.

## Practices and health-seeking behavior of animal bite victims

In our survey, we encountered 84 dog bite victims in 64 households. We investigated the actions taken after the bite incident in each case (n = 81, 3 missing data). Most (80.3%) washed the wound after the bite; 60.5% washed the wound with soap and water, while 21% washed it with water alone. Misconceptions on wound care were still practiced by a minority of the respondents, with 14.8% reporting that they induced bleeding and 4.9% saying they applied garlic to the wound. Almost three-fourths of the bite victims (76.5%) reported that they

**Table 3. Univariate and Multivariate Logistic Regression Analyses of the Factors Associated with High Knowledge Score about Rabies.**

| | Score N (%) | | Univariate analysis | | Multivariate analysis | |
|---|---|---|---|---|---|---|
| Factors | Low score | High score | OR (95% CI) | P-value | Adjusted OR (95% CI)* | P-value |
| Education status of respondents (n = 724) | | | | | | |
| Secondary education | 142 (48.0) | 154 (52.0) | Ref | | Ref | |
| None | 43 (74.1) | 15 (25.9) | 0.32 (0.17–0.60) | <0.01 | 0.30 (0.16–0.59) | <0.01 |
| Primary education | 161 (73.2) | 59 (26.8) | 0.34 (0.23–0.49) | <0.01 | 0.33 (0.22–0.49) | <0.01 |
| Post-secondary education | 81 (53.3) | 71 (46.7) | 0.81 (0.55–1.20) | 0.29 | 0.72 (0.48–1.08) | 0.12 |
| Municipality | | | | | | |
| Barangay A, Calumpit | 85 (66.9) | 42 (33.1) | Ref | | Ref | |
| Barangay B, Pulilan | 70 (59.3) | 48 (40.7) | 1.39 (0.82–2.34) | 0.22 | 1.51 (0.87–2.62) | 0.14 |
| Barangay C, Paombong | 79 (68.1) | 37 (31.9) | 0.95 (0.55–1.62) | 0.85 | 1.11 (0.63–1.97) | 0.72 |
| Barangay D, Bulakan | 73 (57.9) | 53 (42.1) | 1.47 (0.88–2.45) | 0.14 | 1.52 (0.89–2.60) | 0.13 |
| Barangay E, Hagonoy | 68 (55.3) | 55 (44.7) | 1.64 (0.98–2.73) | 0.06 | 1.83 (1.06–3.15) | <0.05 |
| Barangay F, Guiguinto | 52 (44.4) | 65 (55.7) | 2.53 (1.51–4.25) | <0.01 | 2.92 (1.68–5.06) | <0.01 |
| Own a dog and allow it to roam freely | | | | | | |
| No dog | 221 (66.2) | 113 (33.8) | Ref | | Ref | |
| Own at least one dog but not allowed to roam freely | 136 (50.2) | 135 (49.8) | 1.94 (1.40–2.70) | <0.01 | 2.01 (1.41–2.87) | <0.01 |
| Own at least one dog that is allowed to roam freely | 70 (57.4) | 52 (42.6) | 1.45 (0.95–2.22) | 0.09 | 1.84 (1.17–2.92) | <0.01 |

* The final model includes "age" and "sex" related to knowledge level.

CI, confidence interval

consulted a physician or nurse, 12.4% consulted traditional healers, while 13.6% did not go for a consultation (S2 Table).

## Discussion

We conducted a household survey in a densely populated area with the highest number of human rabies cases reported in 2018, using complete household mapping and systematic randomization. We observed a larger owned dog population in the area than was expected when applying the estimates recommended by the NRPCP. We estimated the owned dog population in the selected areas by ascertaining the dog: human ratio and extrapolating it using the latest human census, conducted in 2015. Using these data, the owned dog: human ratio was 1:3.7. Our data show that there might have been a three-fold underestimation of the estimated population (152 484 vs. 56 548) when using the 1:10 ratio. Using provincial vaccination data, we determined that the dog vaccine coverage using the 1:10 ratio was inadequate at 41.7% and more so when using a 1:3.7 ratio (estimated vaccine coverage at 15.5%). Both these values fall below the recommended 70% vaccination coverage of dogs in high-risk areas by the OIE [12]. PVO data is limited to the rabies vaccination campaign by the government and does not include the number of rabies vaccinations outside this campaign. Although we expect that there would be underestimation when using this data, it provides a partial estimate of vaccine coverage. The NRPCP also uses the same data sent by municipal, provincial and city veterinary offices to estimate vaccine coverage. Based on the latest 2019 NRPCP manual of procedures, the total dog vaccination coverage for the Philippines in CY 2018 is 53.30% with a total of 4 758 226 vaccinated dogs [6]. Difficulties encountered during an animal vaccination campaign were elucidated by an older study by Robinson et al., which cites the following difficulties; dogs could not be restrained, owners were not at home, and fear of injury resulting from vaccination [14]. Table 4 summarizes other studies in the Philippines in which the owned dog population and the total dog population were estimated in various areas of the country. In some studies, a purposive sampling methodology was adopted in which dog- or pet-owning households were surveyed [13,15], while in others, a sampling technique that combined random and systematic methods was undertaken [14,16]. This combined random and systematic method is also commonly described in other studies conducted in Africa [28–30]. Beran et al. conducted the earliest known study on a dog population in the Philippines in 1982 and reported a dog: human ratio between 1:6 and 1:8. This study, however, was quite limited, with only 64 dog-owning households surveyed in Dumaguete [13]. In 2015, Bernales and Basitan conducted a study that included 1200 dog-owning households in the Bicol region and reported a dog: human ratio of 2:1 [15]. Although both studies provide insight into the dog population, they present an inflated estimate because a purposive sampling method was used to select the dog-owning households in both. We hypothesized that the results of studies that include both households that own and do not own dogs may more closely represent the actual owned dog population. The earliest study we discovered using a cluster survey of both dog-owning and non-dog-owning households was conducted by Robinson et al. in 1996, and it investigated 210 households in predominantly rural areas of Sorsogon. They reported a dog: human ratio of 1:3.8 [14]. The study conducted by Valenzuela et al. included a multi-year dog population survey in Ilocos Norte in which they reported that the dog: human ratio was 1:3.8 in 2014 [17]. Two investigations led by the Humane Society International reported a ratio of 1:3.9 at Quezon City, a highly urbanized city located in the national capital in 2016 [21], and in rural and urban areas of Cebu City, with a ratio of 1:5.5 in 2017 [16]. The only study that focused on the unowned dog population was conducted by Valenzuela et al. in rural and urban areas of Ilocos Norte in 2016. The dog population was estimated using human density data from the Landscan

**Table 4. Summary of Dog: Human ratios From Studies Conducted in the Philippines.**

| Authors | Year | Sampling Method | Sample size | Dog and human population | Location | Urban or rural | Dog: human ratio |
|---|---|---|---|---|---|---|---|
| Beran et al. [13] | 1982 | Household survey (Only dog-owning households) | 64 households | Not specified | Dumaguete | Both rural and urban | 1:6–8* |
| Robinson et al. [14] | 1996 | Cluster survey (Dog-owning and non-dog-owning households) | 210 households | 297 owned dogs, 1131 humans | Sorsogon | Predominantly rural | 1:3.8 |
| Bernales and Basitan [15] | 2015 | Dog owners, purposive sampling (Only dog-owning households) | 1200 dog owners | 2193 owned dogs, 1200 households | Bicol Region (Camarines Norte, Camarines Sur, Albay and Masbate) | Not specified | 2:1* (dog: household ratio) |
| Amano [21] | 2016 | Systematic random sampling (Dog-owning and non-dog-owning households) | 950 households | 177 289 owned dogs, 699 348 humans | Quezon City | Urban | 1:3.9 |
| Amano and Kartal [16] | 2017 | Systematic random sampling (Dog-owning and non-dog-owning households) | 2020 households | 167 293 owned dogs, 922 611 humans | Cebu City | Both rural and urban | 1:5.5 |
| Valenzuela et al.[17] | 2014 | Cluster sampling (Dog-owning and non-dog-owning households) | Not specified | 149 748 owned dogs | Ilocos Norte | Both rural and urban | 1:3.8 |
| | 2016 | Cluster sampling (Dog-owning and non-dog-owning households) | Not specified | 278 691 dogs (household and freely roaming), 593 081 humans | Ilocos Norte | Both rural and urban | 1:2.24** |
| Current study | 2019 | Systematic random sampling with complete household mapping (Dog-owning and non-dog-owning households) | 727 households | 878 dogs, 3256 humans, | Bulacan | Both rural and urban | 1:3.7 |

* Dog: human ratio is high because the study only surveyed dog-owning households.

**Ratio including the estimated unowned dog population

data of the Oak Ridge National Laboratory. They reported a dog: human ratio of 1:2.24 [17]. The ratios reported previously are not far removed from the estimate in this study, which is 1:3.7. All the other studies reported on the owned dog: human ratio, except the study done by Valenzuela which tried to estimate the total dog: human ratio using human density data.

Therefore, there is an urgent need to replace the ratio of 1:10 in the NRPCP Manual of Procedures to estimate the dog population. Improving evaluation techniques and supporting regular dog population studies in different areas of the country would help refine the dog population estimates and support the need to procure more animal rabies vaccines to ensure better coverage.

Urban and suburban areas in the Philippines witness rapidly increasing population densities and overcrowded households [31]. The methodologies described in previous studies may cause selection bias because many small households are not easily accessible, and a household register is not available in many communities. To the best of our knowledge, complete identification of households or systematic sampling has not been undertaken in any study so far. In addition, it is essential to determine the precise dog population in areas where rabies incidence is high. This study is unique among its peers as our method included the complete mapping of every household in the barangay that enabled us to register 9173 households, of which 727 households were systematically selected for the survey.

## Rabies knowledge and practices

Knowledge regarding rabies was relatively good in the surveyed barangays, with 85.4% of households familiar with the disease. These results are similar to those reported by Amparo

et al.; this study reported that over 80% of the respondents were aware of the nearest ABTCs [7]. Specific rabies knowledge that is strong in the community includes species vulnerable to rabies, common signs of rabies, and knowledge regarding the nearest ABTC.

More importantly, our study identified limitations in the community's knowledge of rabies. The knowledge that needs improvement primarily revolves around how humans and animals get rabies. More than half of the respondents (57.2%) erroneously answered that humans could get rabies when dogs bite them, while 20.4% answered correctly that rabid dogs could transmit rabies. Almost a quarter of the respondents said dogs could get rabies by eating contaminated food (20.36%). Although it seems like the association between dog bites and rabies is well established, the understanding that rabies is a disease affecting dogs and that only dogs with rabies can transmit the disease to humans and other dogs is lacking. This leads to the perception that rabies is innate in dogs and is not considered a disease that can be eliminated. Addressing this incorrect impression of rabies is an essential step in eliminating the disease. A targeted education campaign would not only explain the dangers of rabies but would also focus on aspects where there is insufficient knowledge. Our logistic regression analysis showed a significant difference in rabies knowledge depending on education level, barangay, and dog ownership. We noted that residents of households with dogs scored significantly higher than those without dogs.

Our findings also show good practices and health-seeking behaviors among bite victims, with most administering the correct first aid and seeking medical consultation. This contrasts with the findings presented by Deray and Amparo, which showed poor health-seeking behavior (HSB) among school-age children and the general population, respectively [7,32]. It was also noted that a minority of the participants still consulted with a traditional healer or did not seek any consultations.

## Limitations

Our population estimates are limited to the owned dog population. We did not include the stray dog population in our survey because of limitations in time and uncertainty on the methodology for determining their numbers. A systematic review of 26 surveys including freely roaming dogs revealed several issues concerning the methods used and concluded that there is a high prevalence of estimates with low validity [33]. However, we recognize the importance of estimating the freely roaming dog population and recommend that future dog population surveys consider including them with improved methodology and implementation design to address the issues raised. The significance of including this parameter is highlighted by the findings of a study in Ilocos Norte, wherein a multi-year dog population survey was conducted. Including the estimates of the stray dog population raised their ratio from 1:3.8 to 1:2.24 [17]. However, it should also be argued that an effective dog population control campaign and stricter implementation of the anti-rabies law, with penalties associated with allowing unrestrained dogs to roam freely would reduce the problem of the stray dog population. Our vaccine coverage estimate is based on the data given by the provincial veterinary office. This data is limited to the government led animal rabies vaccination efforts and does not include vaccination outside this program. It is expected that by using this information, vaccine coverage would be underestimated. However, this was the only available data at the time of study and provides a partial measure of vaccine coverage. Currently, there is no available data on the percentage of dogs vaccinated by the program and those vaccinated by veterinarians in their practices.

We also discovered a significant difference in knowledge between the barangays, although only six were included. Inclusion of more barangays could better represent the province,

particularly regarding differences between urban and rural communities. Our data on rabies practices and HSB must be interpreted with caution because it may not represent the whole population as the sample size is limited. Moreover, our study has limited responses concerning rabies practices and HSB because we limited the time frame to one year, and only incidents of animal bites in 2019 were included. This was done to reduce the possible effects of recall bias. However, this led to a low response rate because the household history of animal bites during that time was lower than expected compared to what was reported by Bernales and Basitan of 20.7% [15].

## Conclusions

Our results, and previous studies, indicate that using the 1:10 dog: human ratio as a basis for the dog population in vaccination campaigns might result in an underestimation which can result in inadequate animal vaccination coverage. Therefore, we recommend the revision of the dog: human ratio estimated by the NRPCP. Further, although the communities we surveyed were relatively knowledgeable about rabies, we identified areas for improvement in rabies knowledge, particularly in how humans and animals get rabies. Consequently, we propose content analysis of the existing rabies health education materials to identify over-and under-emphasized aspects and to improve information, education, and communication in the future.

## Supporting information

**S1 Table. Scoring system.**
(DOCX)

**S2 Table. Practices of households with bite victims ($n$ = 81, 64 households).**
(DOCX)

**S3 Table. Number of human and animal rabies cases in Bulacan 2015–2018 (Department of Health–Region 3 Office, Department of Agriculture–Region 3 Office).**
(DOCX)

**S1 File. Questionnaire in XLSForm format, English and Filipino.**
(XLSX)

## Acknowledgments

We want to acknowledge the contributions of Dr. Mary Elizabeth Miranda, Dr. Cristina Ambas, Ms. Angela Kae Chang, Ms. Maryfrance Macaldo, Ms. Madel Line Dela Cruz, and Ms. Mary Rose Pelingon, during protocol development. We want to extend our special thanks to Mr. Koji Shimokawa and Ms. Annabelle La Valle for their efforts in project management for the JAPHOR project. We would also like to thank the Provincial Government of Bulacan, its staff from the Provincial Health Office, the Provincial Veterinary Office, and the local barangay officials who supported the conduct of the study.

## Author Contributions

**Conceptualization:** Timothy John R. Dizon, Nobuo Saito, Marianette Inobaya, Alvin Tan, Mark Donald C. Reñosa, Maria Glofezita O. Lagayan, Kazunori Kimitsuki, Akira Nishizono, Beatriz P. Quiambao.

**Data curation:** Timothy John R. Dizon, Nobuo Saito, Jerric Rhazel Guevarra.

**Formal analysis:** Timothy John R. Dizon, Nobuo Saito, Yasuhiko Kamiya.

**Funding acquisition:** Nobuo Saito, Akira Nishizono, Beatriz P. Quiambao.

**Investigation:** Timothy John R. Dizon, Nobuo Saito, Micah Angela O. Salunga, Patricia Mae T. Lacanilao, Maria Glofezita O. Lagayan, Kazunori Kimitsuki, Akira Nishizono.

**Methodology:** Timothy John R. Dizon, Nobuo Saito, Marianette Inobaya, Alvin Tan.

**Project administration:** Timothy John R. Dizon, Nobuo Saito, Mark Donald C. Reñosa, Thea Andrea Bravo, Vivienne Endoma, Catherine Silvestre, Micah Angela O. Salunga, Patricia Mae T. Lacanilao.

**Resources:** Timothy John R. Dizon, Nobuo Saito, Micah Angela O. Salunga, Patricia Mae T. Lacanilao, Beatriz P. Quiambao.

**Software:** Timothy John R. Dizon, Nobuo Saito, Jerric Rhazel Guevarra.

**Supervision:** Timothy John R. Dizon, Mark Donald C. Reñosa, Thea Andrea Bravo, Vivienne Endoma, Catherine Silvestre, Micah Angela O. Salunga, Patricia Mae T. Lacanilao, Beatriz P. Quiambao.

**Validation:** Timothy John R. Dizon, Marianette Inobaya, Micah Angela O. Salunga, Patricia Mae T. Lacanilao.

**Visualization:** Nobuo Saito.

**Writing – original draft:** Timothy John R. Dizon, Nobuo Saito, Yasuhiko Kamiya.

**Writing – review & editing:** Timothy John R. Dizon, Nobuo Saito, Marianette Inobaya, Alvin Tan, Mark Donald C. Reñosa, Thea Andrea Bravo, Vivienne Endoma, Catherine Silvestre, Jerric Rhazel Guevarra, Yasuhiko Kamiya, Maria Glofezita O. Lagayan, Kazunori Kimitsuki, Akira Nishizono, Beatriz P. Quiambao.

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
