## [Decision Letter · Decision Letter 0]

14 Jul 2021

Dear Dr. Dizon,

Thank you very much for submitting your manuscript "Identifying gaps in household dog population and rabies knowledge in selected municipalities in Bulacan, Philippines: a cross-sectional study" for consideration at PLOS Neglected Tropical Diseases. As with all papers reviewed by the journal, your manuscript was reviewed by members of the editorial board and by several independent reviewers. In light of the reviews (below this email), we would like to invite the resubmission of a revised version that takes into account the reviewers' comments. 

We cannot make any decision about publication until we have seen the revised manuscript and your response to the reviewers' comments. Your revised manuscript is also likely to be sent to reviewers for further evaluation.

Sincerely,

José Reck Jr.

Associate Editor

Victoria Brookes

Deputy Editor

EDITOR COMMENTS:

That´'s an interesting work, important to the field. It will be improved after authors fulfill referees' suggestions.

Reviewer's Responses to Questions

**Key Review Criteria Required for Acceptance?**

**Methods**

-Are the objectives of the study clearly articulated with a clear testable hypothesis stated?

-Is the study design appropriate to address the stated objectives?

-Is the population clearly described and appropriate for the hypothesis being tested?

-Is the sample size sufficient to ensure adequate power to address the hypothesis being tested?

-Were correct statistical analysis used to support conclusions?

-Are there concerns about ethical or regulatory requirements being met?

Reviewer #1: See the attached PDF. The objectives of the study need to be more clearly articulated.

Reviewer #2: The authors conducted a well elaborate survey, focused upon precise household mapping and systematic sampling method randomized, avoiding selection bias that took place in previous studies. Despite the no evidenced conclusion relating the non-achievement of the ideal vaccination coverage to incorrect dog: human ratio, and the rather small temporal and spatial extent of the study, the paper brings valuable information, suggestions and solutions, and it should make, after minor revision, a relevant contribution to policy makers in the field of rabies control in the Philippines.

**Results**

-Does the analysis presented match the analysis plan?

-Are the results clearly and completely presented?

-Are the figures (Tables, Images) of sufficient quality for clarity?

Reviewer #1: See the attached PDF. The results do follow the analysis plan but could benefit from increase clarity.

Reviewer #2: Detailed comments: 

Line 213

Is 2018 the only data available? Is there a trend in dog vaccination coverage over the years? How is it?

Line 224

I suggest converting to US$ to facilitate understanding and comparison between currency

**Conclusions**

-Are the conclusions supported by the data presented?

-Are the limitations of analysis clearly described?

-Do the authors discuss how these data can be helpful to advance our understanding of the topic under study?

-Is public health relevance addressed?

Reviewer #1: The limitation are clearly described, the conclusions are supported by the data. Overall, the data will advance our understanding of rabies, in particular regarding dog:human ratios.

Reviewer #2: Although the authors bring the inadequate ratio and the lack of a precise dog population as an impairment to achieve the recommended coverage in dog vaccination, no elements are supporting that filling this gap will improve the coverage. Explanation about the previous dog vaccination campaigns is missing. 

With modifications addressing the detailed comments and adding some elements about the previous data , this will be a worthwhile paper addressing a relevant public health matter of concern.

Detailed comments:

Line 323

Even with the current ratio, the coverage is far below. What are the difficulties alleged?

Line 393

There is a correlation, but the information that using the 1:10 dog:human ratio as a basis for vaccination targets will result in an inadequate vaccination coverage is not supported by the data here presented, I suggest rephrase

**Editorial and Data Presentation Modifications?**

Reviewer #1: See the attached PDF.

Reviewer #2: More information about the difficulties faced during previous campaigns of dog vaccination would allow a more consistent analysis. For example, when the massive animal vaccination happens, Is there a shortage of canine vaccines? Additionally, more information about the occurrence of human rabies in the Philippines (a table with the historical series would be enough) will provide a basic understanding of the local situation.

Detailed comments:

Line 28

What is “Last few years”? Five years? Decade?

Line 29

For a good summarizing and quick identification whether the paper meets what readers are looking for, I suggest providing a quick definition (such as neighbourhoods) for "barangays"

Line 30

In what period? The highest in the last decade? Five years?

Line 66

Suggest detailing in “free of canine (dog) rabies”

Line 77

Doesn’t make sense. How does PEP in humans impact rabies in dogs? What is this "little impact"?

Line 80

Carrying out the vaccination without the precise number is a challenge, but it is possible to be done. For example, developing countries in America which also did not have an accurate estimate of their canine population and managed to drastically reduce/eliminate dog-related rabies. Suggest rephrase

Line 87

The authors cite several studies about dog population, but the recommendation of the national rabies committee has not been changed because of insufficient data from the dog population survey. Is there a reference from the national rabies committee citing the insufficient data? The previous studies were considered by the national rabies committee?

Line 115

This phrase is not clear. How many cases? 32 cases and also 159 cases in the same province? These 159 (2014 to 18) comprise the 32? The electronic site, cite as reference, with the manual of the Philippines National Rabies Prevention and Control Program it is not working

Line 131

What is Northern Samar? A Province?

Line 157

Were those who had never heard of rabies also counted to the knowledge and perception of the respondents regarding rabies?

**Summary and General Comments**

Reviewer #1: (No Response)

Reviewer #2: The main strengths of this paper are that it addresses, and identify gaps, in information that are pivotal for rabies control:dog population, and knowledge about rabies in local population. The authors objectively identified limitations in the community’s knowledge of rabies, providing specific targets for educational campaigns and also informing which sources of information were more efficient and those that remained underutilized.

PLOS authors have the option to publish the peer review history of their article (what does this mean?). If published, this will include your full peer review and any attached files.

Reviewer #1: No

Reviewer #2: Yes: Aline A Scarpellini Campos
---

## [Decision Letter · Decision Letter 1]

25 Oct 2021

Dear Dr. Dizon,

We are pleased to inform you that your manuscript 'Household survey on owned dog population and rabies knowledge in selected municipalities in Bulacan, Philippines: a cross-sectional study' has been provisionally accepted for publication in PLOS Neglected Tropical Diseases.

Best regards,

José Reck Jr.

Associate Editor

Victoria Brookes

Deputy Editor

Reviewer's Responses to Questions

**Key Review Criteria Required for Acceptance?**

**Methods**

-Are the objectives of the study clearly articulated with a clear testable hypothesis stated?

-Is the study design appropriate to address the stated objectives?

-Is the population clearly described and appropriate for the hypothesis being tested?

-Is the sample size sufficient to ensure adequate power to address the hypothesis being tested?

-Were correct statistical analysis used to support conclusions?

-Are there concerns about ethical or regulatory requirements being met?

Reviewer #2: The authors addressed all my comments. No restrictions/observations to the current text

**Results**

-Does the analysis presented match the analysis plan?

-Are the results clearly and completely presented?

-Are the figures (Tables, Images) of sufficient quality for clarity?

Reviewer #2: The authors addressed all my comments. No restrictions/observations to the current text

**Conclusions**

-Are the conclusions supported by the data presented?

-Are the limitations of analysis clearly described?

-Do the authors discuss how these data can be helpful to advance our understanding of the topic under study?

-Is public health relevance addressed?

Reviewer #2: The authors addressed all my comments. No restrictions/observations to the current text

**Editorial and Data Presentation Modifications?**

Reviewer #2: The authors addressed all my comments. No restrictions/observations to the current text

**Summary and General Comments**

Reviewer #2: The authors addressed all my comments. No restrictions/observations to the current text

PLOS authors have the option to publish the peer review history of their article (what does this mean?). If published, this will include your full peer review and any attached files.

Reviewer #2: No

---

## [Editor Report · Acceptance letter]

13 Jan 2022

Dear Dr. Dizon,

We are delighted to inform you that your manuscript, "Household survey on owned dog population and rabies knowledge in selected municipalities in Bulacan, Philippines: a cross-sectional study," has been formally accepted for publication in PLOS Neglected Tropical Diseases.

Best regards,

Shaden Kamhawi

co-Editor-in-Chief

Paul Brindley

co-Editor-in-Chief
